# Theory of Mind for Multi-Agent Collaboration via Large Language Models

**Huao Li[1], Yu Quan Chong[2], Simon Stepputtis[2], Joseph Campbell[2],**
**Dana Hughes[2], Michael Lewis[1], Katia Sycara[2]**

[1] University of Pittsburgh, Pittsburgh, PA
`hul52,cmlewis@pitt.edu`
[2] Carnegie Mellon University, Pittsburgh, PA
`yuquanc,sstepput,jacampbe,danahugh,sycara@andrew.cmu.edu`

## Abstract

While Large Language Models (LLMs) have demonstrated impressive accomplishments in both reasoning and planning, their abilities in multi-agent collaborations remains largely unexplored. This study evaluates LLM-based agents in a multi-agent cooperative text game with Theory of Mind (ToM) inference tasks, comparing their performance with Multi-Agent Reinforcement Learning (MARL) and planning-based baselines. We observed evidence of emergent collaborative behaviors and high-order Theory of Mind capabilities among LLM-based agents. Our results reveal limitations in LLM-based agents' planning optimization due to systematic failures in managing long-horizon contexts and hallucination about the task state. We explore the use of explicit belief state representations to mitigate these issues, finding that it enhances task performance and the accuracy of ToM inferences for LLM-based agents.

## 1 Introduction

Recent large language models (LLMs), such as GPT-4 (OpenAI, 2023), have demonstrated impressive competencies across a wide array of domains and tasks, ranging from mathematics to law, without the need for fine-tuning or special prompting (Bubeck et al., 2023). This advancement has significantly transformed the landscape of Natural Language Processing (NLP) research. Instead of developing domain-specific models for downstream applications, focus has shifted towards evaluating and harnessing LLMs' abilities to solve novel tasks. Such a shift is consistent with the idea of studying machine behaviors, an interdisciplinary approach that expands the conventional bounds of computer science and integrates insights from diverse scientific fields (Rahwan et al., 2019). Drawing inspiration from team science and group psychology (Hagendorff, 2023), our study concentrates on collective machine behavior, evaluating

LLMs' proficiency in multi-agent collaborations.

There is ongoing debate regarding the intelligence levels of modern LLMs. While some argue that LLMs excel primarily in linguistic competence and struggle with cognitive abilities beyond language, known as functional competence, others demonstrate that LLMs can exhibit cognitive skills such as formal reasoning and world knowledge comprehension (Mahowald et al., 2023; Bubeck et al., 2023). Motivated to explore this argument, we designed a text-based game to evaluate LLMs' ability in embodied interactions, including exploring unknown environments, maintaining beliefs about the world and collaborating with other agents, which is critical for natural social interactions and artificial general intelligence (AGI).

Theory of Mind, the capacity to reason about others' concealed mental states, is fundamental to human social interactions, collaborations, and communications (Zhang et al., 2012). As LLMs increasingly participate in diverse social interactions with humans, their social intelligence is expected to improve for them to become effective collaborators (Williams et al., 2022; Li et al., 2022). For instance, a proficient AI assistant should be able to infer a human's preferences based on previous experiences without needing to ask. Recent studies have applied classic Theory-of-Mind tasks to several LLMs, concluding that current models (e.g., GPT-4) perform comparably to 9-year-old children (Kosinski, 2023). However, the research community has expressed doubts about the validity of text-based ToM tests on machine intelligence(Ullman, 2023; Sap et al., 2023). In response, our study proposes a novel evaluation of LLMs' high-order ToM in interactive teamwork scenarios, encompassing dynamic belief state evolution and rich intent communication between multiple agents.

The main contributions of this study include that we:

- Evaluate LLM-based agents' embodied interaction capability in multi-agent collaborative tasks against reinforcement learning and planning-based baselines

- Identify systematic failures that limit the collaboration efficiency of LLM-based agents, and propose a prompt-engineering method to mitigate those failures by incorporating explicit belief state representations about world knowledge in the model input

- Propose a novel evaluation of LLMs' high-order ToM in interactive teamwork scenarios, encompassing dynamic belief state evolution and rich intent communication between multiple agents

## 2 Related Work

### 2.1 Large language models

Large language models, trained on vast text corpora, excel in text completion and various other Natural Language Processing (NLP) applications (Chowdhery et al., 2022; Thoppilan et al., 2022). Recent studies highlight their abilities for reasoning (Bubeck et al., 2023; Wei et al., 2022) and action plan generation (Liu et al., 2023; Xie et al., 2023), particularly when utilizing prompt engineering techniques like chain-of-thought. However, some researchers note these models' limitations in forming actionable plans when interacting with real-world objects (Ahn et al., 2022; Huang et al., 2022). GPT-4's capacity for embodied interactions via text-based games and real-world problems was assessed by Bubeck et al. (2023). Further studies explored the potential of LLM-powered embodied agents in Minecraft (Wang et al., 2023b,a). These investigations suggest that LLMs can perform tasks requiring environment understanding, task comprehension, action planning, feedback interpretation, and subsequent adaptation. Our study seeks to broaden this understanding by evaluating LLMs' planning abilities in cooperative multi-agent scenarios.

### 2.2 Theory of Mind

Prior research has tested LLMs' Theory of Mind (ToM) via variants of text-based tests such as the unexpected transfer task (also known as Smarties Task) or unexpected contents task (also known as the "Maxi Task" or "Sally–Anne" Test) (Kosinski, 2023; Moghaddam and Honey, 2023). Results indicate that leading LLMs can pass more than 90% of these test cases. In contrast, Ullman (2023) found that LLMs struggle with complex ToM inferences involving communication or second-order beliefs. In our study, ToM evaluations occur in the midst of an interactive team task, where the mental states of agents change dynamically with each interaction. As agents exchange information through communication at every timestamp, the complexity of reasoning increases, since agents' mental states may be updated through both observations and communication. Thus, our tests can be considered more challenging than the static text-based tests used in prior research.

Theory of Mind (ToM) has been employed to enhance the performance of artificial agents in various contexts. Lim et al. (2020) introduced a method to integrate Bayesian Theory of Mind (BToM) (Baker et al., 2017) with optimal-planning agents in a cooperative game. The results indicate that an explicit representation of others' intentions enhances the performance of both agent-only and human-agent teams. SymbolicToM allows language models to maintain an explicit symbolic ToM for multiple characters in reading comprehension tasks using graphical representations (Sclar et al., 2023). Moreover, there is a significant body of research focusing on the application of ToM to boost collaboration in multi-agent reinforcement learning (Oguntola et al., 2023; Yuan et al., 2021). Inspired by these prior studies, we aim to enhance LLM-based agents' collaborative behaviors through explicit belief representations.

### 2.3 Multi-agent collaboration

Team science researchers have studied human collaborative behaviors for decades, covering topics such as leadership, communication, team dynamics, team cohesion, and shared situation awareness (Riedl et al., 2021). However, the transferability of these findings to hybrid human-agent teams or fully automated teams remains largely unexplored. Park et al. (2023) utilized ChatGPT to operate a sandbox environment populated by generative agents, observing emergent social behaviors among LLM-based agents. That study primarily focused on the feasibility of running such a sandbox environment with LLMs, rather than specifically on the collaborative behaviors of machine intelligence.

## 3 Multi-agent Collaboration Tasks

To evaluate the capability of LLM-based embodied agents, we design a multi-agent environment to simulate the collaborative and problem-solving dynamics of a search and rescue mission.

### 3.1 Task environment

3 agents (i.e. Alpha, Bravo, and Charlie) emulate specialists in a team, with the objective to locate and safely defuse color-coded bombs scattered in an unexplored environment. Each bomb exhibits unique phase sequences in $m$ colors, requiring the correct order of wire cutters for defusing. Team members start with different colored cutters and must coordinate and synchronize efforts for efficiency. The environment is conceptualized as a connected graph, with $n$ nodes representing $n$ rooms linked by several edges symbolizing hallways. In each round, the agents can choose from three classes of actions: moving to one of the $n$ rooms, inspecting a bomb's phase sequence in the current room, or using one of the $m$ wire-cutters. The size of action space depends on the problem scale (i.e. $n + m + 1$). Agents' observation are limited to their current room's contents and agent status. They are updated periodically about team scores, current room contents, teammates' locations and available tools. The team is rewarded $10*x$ points when a $x$-phase bomb is successfully defused.

The evaluation environment comprises five rooms ($n = 5$) and five bombs, including two single-phase, two double-phase, and one triple-phase bombs. Bomb stages might have three different colors ($m = 3$). Each successfully defused bomb awards the team 10 points per processed phase, resulting in 90 as the maximum score per mission. Team performance is measured using two metrics: the team score, indicating coordination quality, and rounds to completion, measuring collaboration efficiency. A trial concludes when the team has defused all bombs, exceeded the time limit (i.e., 30 rounds), or entered a deadlock by repeating outputs.

### 3.2 Text game interface

The initial task environment is implemented for MARL agents based on gym API (Brockman et al., 2016). To facilitate interaction between LLM-based agents with the environment, we've integrated the task environment with a text interface.

At each round (i.e. timestamp), the team's three agents sequentially interact with the environment, both receiving observations and performing actions via natural language interaction. A built-in communication mechanism enables text message exchange among agents per round. Importantly, agents remain oblivious to each other's actions and outcomes unless communicated, facilitating Theory of Mind inference opportunities.

Specifically, a rule-based text interface translates observations into natural language descriptions and encodes agent chats into abstract action selections. For observations, the text interface extracts state features from the game engine and replaces keywords in the templates. A typical description text includes the current round number, cumulative team score, action feedback, contents of the current room, teammates' locations, and communication messages. Action encoding is done via keyword matching since LLMs are instructed to frame their responses in a certain format and structure. Should an agent produce unintelligible content, such as invalid actions or nonsensical text, the interface provides feedback for error correction. The error messages are generated based on pre-programmed rules and templates, such as "There is no bomb in the current location, Room X, for you to inspect.". Fig. 1 showcases sample interactions between the agent team and task environment via the text interface.

## 4 LLM-based Embodied Agents

We chose to evaluate OpenAI's latest chat completion models, namely gpt-3.5-turbo-0301 and gpt-4-0314, owing to their impressive performance in various benchmarks (Zheng et al., 2023). These models are prompted to engage in a text-based game, with user inputs managed by the above-mentioned game interface. The LLMs functions as embodied agents interacting within the task environment. They are provided with the game's rules as context. For each round, the model is asked to choose actions and communicate messages, based on the current task state observations and past interaction history. Interaction history between the LLM-based agent and text game interface are maintained in the query text until it exceeds the maximum model input size. In our setup, all agents retain memory of the game rules and history from the previous two rounds, amounting to 4096 tokens.

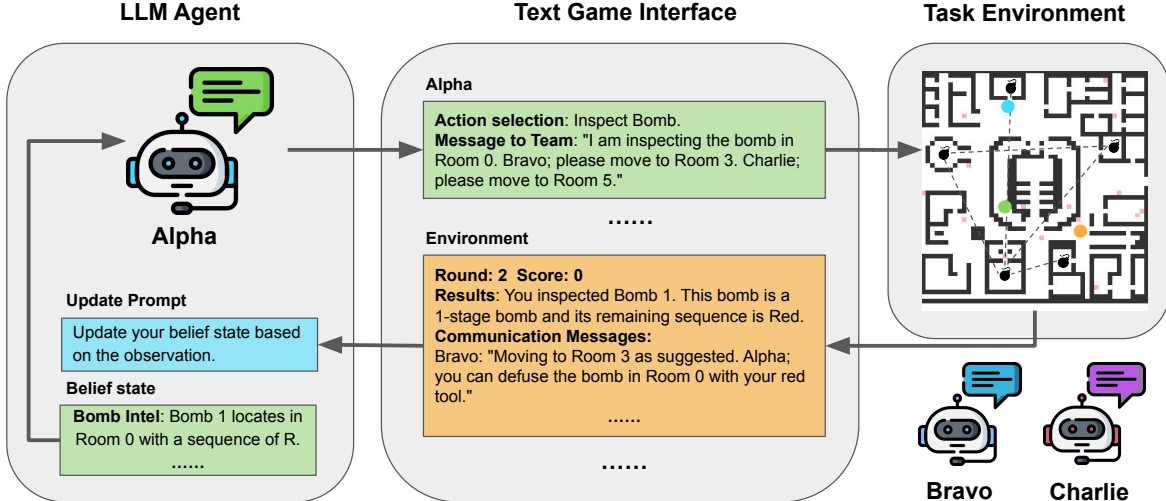

Figure 1: Our proposed framework consist of 3 LLM-based agents, a text game interface and the actual task environment. The natural language outputs of LLM are encoded into abstract actions by the text interface and then sent to task environment. The task environment then processes agent actions and returns observations via the text interface. Upon receiving environmental observations, LLM-based agents are prompted to update their beliefs and output action selections and messages. 3 agents in the team are coded as Alpha, Bravo, and Charlie and take turns to interact with the interface.

## 4.1 Multi-agent communication

Given the collaborative nature of the task scenarios, inter-agent communication is crucial for achieving effective coordination and teamwork. We implemented a communication channel enabling LLM-based agents to share textual messages within the team. Messages, once sent, are immediately broadcast to all team members and reflected in their subsequent observations. For instance, as depicted in Fig. 1, agent Alpha dispatched messages instructing teammates to separate, followed by feedback from agent Bravo. In practice, since agents alternate in message sending, responses from teammates will appear in the observations of the succeeding round.

## 4.2 Belief state

Due to the model input size limitation, LLM-based agents cannot retain the entire interaction history, yet task dynamics require the team to track key long-term information, such as room contents and bomb sequences. To augment the agents' information retention and enhance collaboration, we propose a method of prompt engineering to represent explicit belief states. As illustrated in Fig. 1, upon receiving environmental observations, agents are prompted to update a textual description storing key task-related beliefs. This updated belief state

is preserved in the interaction history and used in subsequent action planning. For instance, after inspecting bomb 1, agent Alpha updated its belief state about the bomb's sequence from unknown to red, retaining this information until further updates.

The proposed belief state is inspired by the idea of chain-of-thought prompting (Wei et al., 2022), wherein a complex reasoning task is broken down into intermediate steps and introduced to the LLM in a few-shot learning manner. Notably, although an initial belief state description is provided to illustrate the proper format and representations, the update rules are entirely zero-shot, relying solely on the LLM's common sense and mission context.

## 5 Experiments

We systematically ablate LLM-based embodied agents and evaluate them in a collaborative task in teams of three. Two modules are manipulated including LLM models (i.e. GPT-4 or ChatGPT) and belief representation (i.e. with or without belief state) resulting in a total of 4 experimental conditions.

## 5.1 Setups

At the beginning of each experimental trial, we assemble a team of three embodied agents and reset the task environment, randomizing starting loca-

| Agents | Score | Rounds to Completion | Valid action % |
|---|---|---|---|
| ChatGPT | 43± 4.7 | 30.0± 0.0 | 62.5% |
| GPT-4 | 90± 0.0 | 28.3± 2.6 | 71.8% |
| GPT-4 + Belief | 90± 0.0 | 12.3± 2.0 | 86.1% |
| MAPPO | 90± 0.0 | 11.0± 0.0 | N/A |
| CBS Planner | 90± 0.0 | 6.0± 0.0 | N/A |
| Random | 38± 14.7 | 30.0± 0.0 | N/A |

Table 1: Task performance of LLM-based agents and baseline conditions. Score represent the average team score in all experiment trials. Length refers the average number of rounds the team took in completing the task. Percentages of valid action measures the proportion of LLM outputs that can be encoded into actions allowed by the task rules. Numbers after ± are 1 standard deviation.

tions, room connections, bomb distributions, and sequences. Agents then take turns providing action choices and communication messages based on their initial observations. It's important to note that each agent only has a partial observation and its own interaction history, with inter-agent communication being the sole means of information diffusion in this fully decentralized team. For LLM-based agents, we set the model temperature parameter to zero and perform three trials of repeated measurement to ensure result stability. Each trial's duration varies from 5 to 120 minutes, depending on task load and model selection.

## 5.2 Baselines

In addition to LLM-based embodied agents, we also include baselines based on MARL and planning methods. For MARL, we consider Multi-Agent Proximal Policy Optimization (MAPPO) (Yu et al., 2022), which has shown strong performance in environments such as the StarCraft Multi-Agent Challenge (SMAC) (Samvelyan et al., 2019). Our model is based on a stateful actor-critic approach building on recurrent neural networks with shared actor and critic models given agent invariance to improve sample efficiency and memory requirements while avoiding the lazy agent problem (Sunehag et al., 2017). We utilise the default hyperparameters for SMAC to train MAPPO in the environment and evaluate its performance from another fixed distribution of randomly generated environments, recording the average score and episode length as well as their standard deviation. Like the LLM agents, MARL agents are able to observe their teammates' locations. Other than the team reward of 10*$x$ points when a $x$-phase bomb is successfully defused, an additional intermediate reward term is implemented as well, where an agent is given a small positive reward of +1 upon the

application of the correct wirecutter in defusing a phase of a bomb and a small negative reward of −1 when it causes a bomb to explode upon the application of the wrong wirecutter. This reward-shaping term allows the agents to more sample efficiently learn the necessary bomb-defusing skills as compared to the relatively sparser team reward.

In addition, we augment a state-of-the-art Multi-Agent Path-Finding (MAPF) algorithm, Conflict-Based Search (CBS) (Sharon et al., 2015), simultaneously generate task assignments with feasible and collision-free paths for agents that adhere to precedence and temporal constraints in order to maximise a user-defined objective instead of the sum of path costs or makespan. Specifically, the user-defined objective is quantified as the return from a user-defined reward function, which is the team reward of 10*$x$ points when a $x$-phase bomb is successfully defused in the stated task. The planner uses a user-defined heuristic (e.g. sort bombs in ascending order of distance from the agents' starting location) to sort the execution order of the actions for the entire task. The ordered actions are then partitioned using a hyperparameter, the number of actions per subtask, to form a subtask (e.g. the two nearest bombs to the agents' starting location). The actions from the subtask are used to generate possible combinations of assignments to agents. The planner returns a feasible solution for the subtask by resolving precedence and temporal conflicts through the expansion of a binary constraint tree in a best-first manner with respect to the return. The solution for the entire task is then composed of the solutions of the subtask sequentially. By considering the entire task of 5 bombs as a single subtask, the planner can be proven to be complete and optimal with respect to the score.

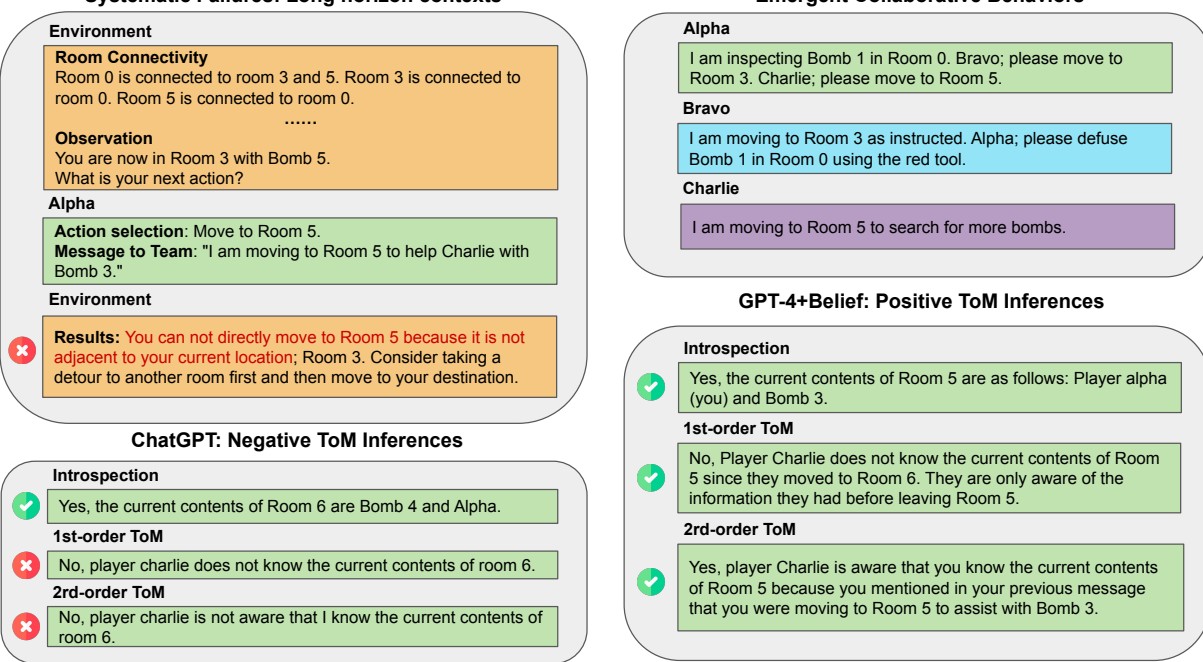

Figure 2: Example interactions between LLM-based agents and the text game interface. The upper left panel showcases one type of systematic failures we observed in LLM's outputs in which long horizon contexts are overlooked. The upper right panel illustrates emergent collaborative behaviors (e.g. emergent leadership) between LLM-based agents. The bottom two panels are quotes of GPT-4+Belief and ChatGPT agents' answers for ToM inference questions.

## 5.3 Theory of mind inferences

Alongside the main task, LLM-based agents are tasked with performing Theory of Mind (ToM) inferences during the mission. These inquiries fall into three categories, aligning with three ToM capability levels. The first category, introspection, assesses an agent's ability to articulate its mental state. The second category, first-order ToM inferences, tests if agents can estimate others' hidden mental states. The third category, second-order ToM inferences, evaluates an agent's ability to infer what others believe about their own mental state.

The design principle of ToM questions is inspired by the Sally–Anne test, the most widely used ToM task in human studies. Every time an agent conducts an action, we pose a belief reasoning question, asking if another agent (i.e., target agent) is aware of the potential consequence of this action. The consequence here can be either a state change (e.g., a bomb has been defused) or a belief change (e.g., Alpha has explored Room 5 and found Bomb 3 in the room). An agent equipped with ToM should realize that while they know the consequence, the target agent might hold a false belief about it. A full list of ToM inference questions can be found in appendix.

To evaluate whether LLM-based agents answer these questions correctly, human annotators were hired to provide subjective judgment based on fully observable interaction and communication history. Specifically, the following standard are considered: 1) if the target agent is present in the current room and observes the consequence, 2) if the target agent has been to this room before, 3) if the consequence has been communicated to the target agent. It is worth mentioning that high-order ToM inferences involving communication are naturally ambiguous. These corner cases were discussed among annotators to ensure a consistent standard across conditions.

## 6 Results

Table 1 and Table 2 present the main experiment results. This section will analyze each metric, examine potential reasons for performance differences, and provide qualitative case studies of experimental trials.

### 6.1 Task performance

Except for the ChatGPT team, all teams manage to defuse all bombs within the time limit. Their efficiency is indicated by the average number of rounds

spent to complete the task. The CBS Planner resolves the task in 6.0 rounds, providing an optimal baseline given its centralized coordination and perfect information sharing. MAPPO, a state-of-the-art multi-agent reinforcement learning algorithm, completes the task in an average of 11.0 rounds after 45 million timesteps of training, serving as a practical baseline.

ChatGPT fails to complete the task in all experiments, averaging a team score of 43.3. On the contrary, teams based on GPT-4 achieve full scores, with those using explicit belief representations being more efficient (28.3 vs. 12.3 rounds). These findings align with previous research demonstrating GPT-4's superior reasoning capabilities compared to ChatGPT (Zheng et al., 2023). LLM-based agents perform exceedingly well in team collaboration tasks, especially considering their fully zero-shot learning and decentralized framework. The incorporation of belief state representation improves team collaboration by reducing invalid actions and enhancing ToM inference capabilities.

## 6.2 Basic embodied interactions

For a successful team, each member should manage individual sub-tasks effectively, a concept known as taskwork in team science (Crawford and Lepine, 2013). This involves understanding task rules, reasoning about action prerequisites and consequences, and interacting with the environment. All LLM-based teams demonstrate basic embodied interaction capabilities, achieving better performance than the random baseline. Additionally, LLM-based agents effectively express their beliefs about task-related information via introspection, as shown in Table 2. All agents show a strong performance (>80%) in understanding world knowledge (e.g., bomb locations) and situation modeling (e.g., interaction history).

## 6.3 Emergent collaborative behaviors

To understand how LLM-based agents match the performance of state-of-the-art MARL methods, we analyzed team trajectories and conducted a qualitative analysis of emergent collaborative behaviors. As shown in the top-right panel of Fig. 2, GPT-4+Belief teams use communication messages to coordinate tasks. Agent Alpha voluntarily takes the role of a team leader, delegating sub-tasks to other members. Other collaborative behaviors common in human teams (Fan and Yen, 2004), such as helping, resolving conflicts, and sharing infor-

mation, also emerge in LLM-based agent teams. These findings suggest that LLMs, through learning from massive language materials, acquire essential teamwork skills without specific collaborative task training.

## 6.4 LLM's systematic failures

However, LLM-based agents' collaboration is less efficient than the optimal baseline. We identify a few systematic failures that LLMs make during team planning and discuss how they impede teamwork progress.

### 6.4.1 Long-horizon contexts

The first bottleneck of LLM-based teams' efficiency is dealing with long-horizon contexts. During the mission, LLMs occasionally output invalid actions that violate task rules, such as moving to non-adjacent rooms or using tools they do not possess. Even though the information about room connectivity and tool allocation are included in the initial prompts and maintained in the inquiry text, LLMs often overlook these details because they are far away from the planning question at the end. The more advanced GPT-4 model performs better in considering long contexts and complex logic, thereby making fewer invalid actions, as shown in Table 1. Our proposed belief state is also helpful in this progress by re-emphasizing task related information in the input prompt.

### 6.4.2 Hallucination

The second type of systematic failure we observe in LLMs is their hallucination about the task state. During the mission, agents might generate valid but infeasible actions, like searching for a defused bomb or claiming the sequence of a bomb without inspection. These actions stem from false beliefs about the game state and do not contribute to task progress. We attribute these hallucinations mainly to the lack of explicit belief representation. Without access to complete interaction history and only partial environment observations, LLM-based agents can't form an accurate belief about the task state. Therefore LLMs might generate imaginations about nonexistent bombs or fake bomb sequences when reasoning about the next action. We evaluate this hypothesis by the GPT-4+Belief condition where LLM-based agents explicitly represent their belief state in text. Results show that the introduction of belief state decreases invalid action by 50.7% and increase the team efficiency by 130%

| Agents | Introspection | 1st ToM | 2rd ToM |
|---|---|---|---|
| ChatGPT | 79.0% | 41.9% | 11.6% |
| GPT-4 | 80.0% | 60.0% | 64.3% |
| GPT-4 + Belief | 97.2% | 80.1% | 69.4% |

Table 2: LLM-based agents' performance in ToM inference tasks. Natural language answers are annotated by experimenters and compared with the ground truth based on global interaction history. Percentages represent the inference accuracy.

### 6.5 Theory of Mind Inference

A critical aspect of teamwork is inferring teammates' mental states, including beliefs, desires, and intentions. We assess LLM-based agents by asking them to conduct Theory of Mind inferences during the mission. As seen in Table 2, LLM-based agents can estimate their own and their teammates' mental states. In the most challenging second-order ToM inference tasks, where agents estimate others' beliefs about their own mental states, GPT-4 + Belief agents correctly respond in nearly 70% of cases. Consistent with team performance, GPT-4 surpasses ChatGPT in all three ToM inference levels, and explicit belief state representation enhances LLM-based agents' ToM capabilities. In the following case study, we'll analyze LLM responses to see how they succeed or fail in certain cases.

### 6.5.1 Case study

As shown in Fig. 2, after Alpha entered Room 5 and observed the contents, we asked whether a teammate in another room (i.e., Charlie) knows Room 5's contents. This is a first-order belief estimation question. GPT-4 answers correctly saying

> "No, Player Charlie does not know the current contents of Room 5 since they moved to Room 6. They are only aware of the information they had before leaving Room 5."

considering both Charlie's current location (not in Room 5) and their interaction history (they've been in Room 5 before). In contrast, ChatGPT fails to consider this history. In the second-order ToM inference case, we asked if Charlie is aware that Alpha knows Room 5's contents. GPT-4+Belief answers correctly by considering previous communications whereas ChatGPT fails.

> "Yes, player Charlie is aware that I know the current contents of Room 5 because I mentioned in my previous message that

I was moving to Room 5 to assist with Bomb 3."

### 6.5.2 Inference under false belief

In some situations, ToM inferences become extremely challenging for LLMs. In our records, this is typically when reasoning involves agent communication. Even with ample context, it's hard for agents to track information transmission while conducting ToM inferences. The most demanding scenarios occur when agents share inaccurate information, leading to false beliefs. This usually happens when LLM-based agents try an invalid action and broadcast false intentions through communication. For instance, when Alpha tells the team they're moving to Room 8 (which is unreachable), and we ask Alpha, "Is Charlie aware that you know the contents of Room 8?" This question is tricky since Alpha failed to reach Room 8. In theory, the most acceptable answer is complex and involves multiple high-order ToM assumptions. For example a human annotated ground truth is:

> "I did not know the contents of Room 8 cause I failed to make it there. But since I told Charlie I will go there, they might assume that I know the contents of Room 8. However, considering Charlie knows my current location, they should be able to tell that I am not able to move to Room 5 and induce that I do not know the contents of Room 5."

In practice, LLM-based agents often grasp only part of the logic or miss the crucial fact that they didn't reach the target room. Quotes from GPT-4+Belief agent:

> "Based on the observation and previous history, Player Charlie is not explicitly aware that I know the current contents of Room 8. However, since I mentioned in my previous message that I was moving to Room 8, Player Charlie might assume

that I would eventually learn about the contents of Room 8."

# 7 Discussions

Our study yields three primary insights. First, Large Language Models (LLMs) demonstrate substantial planning and collaboration capabilities within our task scenarios. With suitable prompt-engineering, teams of LLM-based agents perform comparably to state-of-the-art Multi-Agent Reinforcement Learning (MARL) algorithms. This finding is particularly noteworthy given that MARL agents receive extensive task-specific training with a centralized critic, while LLM-based agents operate in a fully decentralized manner and undertake tasks in a zero-shot setting. Despite prior research highlighting LLMs' limitations in generating actionable plans and interacting with the world, they perform reasonably well when placed in a team and tasked to process actions step-by-step. Particularly, LLMs fine-tuned with Reinforcement Learning from Human Feedback demonstrate emergent social interaction skills in multi-agent collaborations, which might be similar to the collaborative and interactive settings in which human language is primarily learned and used (Sap et al., 2023).

Second, LLMs still fall short of being optimal planners or team players due to systematic failures, such as neglecting long-horizon contexts and making inaccurate assumptions about the task state (a.k.a hallucination). These flaws significantly hinder team collaborations as they can rapidly disseminate misinformation via communication, leading to widespread false beliefs. We attempted to mitigate these issues by allowing LLM-based agents to maintain an explicit belief state about the world. Our findings suggest that modern LLMs can update the given belief descriptions based on their observations, hinting at the potential emergence of advanced cognitive skills such as world knowledge understanding and situation modeling. Moreover, belief state representations offer a structured framework that helps agents track key task-related information, leading to improved team performance.

Finally, our study indicates that the Theory of Mind (ToM) capabilities of LLMs are still limited, particularly when evaluated within interactive teamwork scenarios that involve dynamic belief states and intensive communication. For context, while 5-year-old children can perform second-order ToM inferences (Miller, 2009), adults don't consistently use this ability during communications due to the complexity and ambiguity of social interactions (Keysar et al., 2003). Thus, there's considerable work ahead for LLMs to develop a functional ToM and interact naturally with humans. Our study represents a preliminary effort to devise novel evaluation methods for LLMs' ToM that go beyond traditional tests such as the Sally-Anne test.

# 8 Conclusions

In this study, we assessed the ability of recent large language models (LLMs) to conduct embodied interactions in a team task. Our results demonstrate that LLM-based agents can handle complex multi-agent collaborative tasks at a level comparable with the state-of-the-art reinforcement learning algorithm. We also observed evidence of emergent collaborative behaviors and high-order Theory of Mind capabilities among LLM-based agents. These findings confirm the potential intelligence of LLMs in formal reasoning, world knowledge, situation modeling and social interactions. Furthermore, we discussed two systematic failures that limit the performance of LLM-based agents and proposed a prompt-engineering method that mitigates these failures by incorporating an explicit belief state about world knowledge into the model input.

## Limitations

This study represents an initial effort to understand machine intelligence in complex task scenarios. Several enhancements could improve the experimental setup and offer a more thorough evaluation of LLMs in multi-agent collaborations. First, we could incorporate additional LLMs besides OpenAI's GPT models. As new models emerge with enhanced reasoning capabilities and larger input sizes, their performance in team tasks and ToM inference may also change. Second, the task environment is relatively simple with only five nodes and five bombs. We plan to scale up the environment and introduce more restrictions to test how LLM-based teams react to more challenging tasks. Lastly, the current team consists of three agents with homogeneous policies. It would be intriguing to evaluate how LLM-based agents perform in human-agent teams, especially from a human-centered perspective where issues like trust, transparency, and human-agent co-training can be addressed.

The ToM capability evaluation method used in

this study also has its limitations. Currently, human annotators, who have a global view of the task state and interaction history, generate the ground truth for ToM inference questions. However, this estimation is at best an approximation, assuming agents process information as a rational human would, which might be ambiguous in situations involving false beliefs or miscommunications. A potential alternative could be using each agent's maintained belief state as the ground truth.

The proposed belief state method could extend from introspective belief to first-order or even second-order beliefs. Currently, LLM-based agents maintain a belief state about their own world knowledge in text form. By extending this representation to include other agents' world knowledge, we could equip LLM-based agents with an explicit first-order ToM model. Their ToM capability can be assessed by directly comparing one's first-order belief with another's introspective belief, rather than asking LLMs Sally-Anne style questions.

## 9 Acknowledgements

This work was supported by DARPA award HR001120C0036 and AFOSR award FA9550-18-1-0097.

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

# Appendix

## A  Prompts

### A.1  Task context

Welcome to our interactive text game! In this game, you'll assume the role of a specialist on a search and rescue team. Alongside two other players, you'll navigate a five-room environment with a mission to defuse five hidden bombs.

**The Map**: Imagine a network of rooms represented by a connected graph where each node corresponds to a room, and the edges between nodes depict hallways. The rooms are numbered 0, 3, 6, 5, and 8. Room 0 is connected to all other rooms. Room 5 shares a hallway with room 6. Room 3 is linked to room 8. And room 8 is also connected with room 6. You can only travel to adjacent, directly connected rooms at each turn.

**The Challenge**: Scattered among these rooms are five bombs, each coded with different phases represented by colors. To defuse them, you'll need to use the correct wire-cutting tools in the correct sequence. There are one-phase, two-phase, and three-phase bombs, needing 1, 2, or 3 color-coded tool applications in sequence to disarm. For instance, a bomb with a red-green phase sequence requires the red tool first, then the green one. Points are awarded based on the number of tools used for defusing a bomb, with each tool use worth 10 points. Your task is to maximize the team score as soon as possible. The challenge is that the bomb locations and sequences are unknown to players at the start.

**Tools**: Each player is equipped with two color-coded wire cutters. As player Alpha, you have red and green tools, player Bravo wields green and blue, and player Charlie possesses blue and red.

**Actions**: Each round, you can opt to do one of the following: 1) Move to an adjacent room, 2) Inspect a bomb's phase sequence in your current room, or 3) Apply your wire cutters to a bomb in the current room.

**Communications**: In addition to selecting an action to take from the above list, you can also send communication message texts to both of your teammates in each round. The message text you sent will be shared with both of your teammates in their observation in the next round.

**Observation**: While you can only see what's in your current room and read text messages from teammates. You'll also be informed of the current round number, team score and the current location of your teammates. Your teammates have the same observability as you. They will not be able to know your action and its consequences unless you explicitly communicate.

To facilitate our interaction, reply your action selection and communication messages in this fixed format: Action selection: Your action. Message to Team: "Your Message". To move to an adjacent room, say: 'Move to Room X'. To inspect the sequence of a bomb in your current room, say: 'Inspect Bomb'. To apply a wire cutter tool, say: 'Apply X Tool'. Remember, your replies must adhere strictly to these rules. Feel free to ask clarifying questions if needed. I'll supply the necessary information as we progress. Are you ready to take on this explosive challenge?

### A.2  Initial belief state

Below is your current belief about game state based on your previous observations about the environment and interactions with your teammates. **Your role**: You are playing as Player <agent id>.
**Current round**: 1
**Total team score**: 0.
**Observation:** You are currently in Room 0 with both of your teammates. In the room you also found bomb 1 with unknown sequence. There is no other bomb in the current room.
**Teammate Locations:** Player alpha is in Room 0; Player bravo is in Room 0; Player charlie is in Room 0.
**Room connectivity**:

- Room 0 is connected to room 3, 5, 6, 8

- Room 3 is connected to room 0

- Room 5 is connected to room 0 and 6

- Room 8 is connected to room 0 and 6

**Bomb Intel**:

- Bomb 1: Located in Room 0. The phase sequence is Unknown.

- Bomb 2: Details currently unknown.

- Bomb 3: Details currently unknown.

- Bomb 4: Details currently unknown.

- Bomb 5: Details currently unknown.

**Tool inventory:**

- Alpha: Equipped with red and green wire cutters.

- Bravo: Equipped with green and blue wire cutters.

- Charlie: Equipped with red and blue wire cutters.

**Available action options:**

- To move to an adjacent room, say: 'Move to Room X'.

- To inspect the sequence of a bomb in your current room, say: 'Inspect Bomb'.

- To apply a wire cutter tool, say: 'Apply X Tool'.

- To send a message to your teammates, say: 'Message to Team: "Your Message"'.

## B Environment feedback for Error correction

- Your action is invalid.

- You can not directly move to Room $roomid$ because it is not adjacent to your current location, Room $currentroom$. Consider taking a detour to another room first and then move to your destination.

- There is no bomb in the current current location, Room $currentroom$, for you to inspect.

- You can not apply Tool $toolcolor$ to Bomb $boomid$ because the sequence of this bomb is sequence. You will need to apply other color tool first.

- There is no bomb in your current location, room $roomid$, for you to defuse.

- You do not have Tool $toolcolor$. Consider asking your teammates who have this tool to help you defuse the bomb.

## C Theory of Mind Questions

### C.1 Introspection

- Do you know the current contents of room $roomid$?

- Do you know the state and remaining sequence of bomb $bombid$ has been changed?

- Do you know a bomb phase has just been defused?

- Do you know the sequence of bomb $bombid$?

### C.2 First-order ToM

- Does player $playerid$ know the current contents of room $roomid$?

- Does player $playerid$ know the state and remaining sequence of bomb $bombid$ has been changed?

- Does player $playerid$ know a bomb phase has just been defused?

- Does player $playerid$ know the sequence of bomb $bombid$?

### C.3 Second-order ToM

- Based on the observation and previous history, is player $playerid$ aware of the fact that you know the current contents of room $roomid$?

- Based on the observation and previous history, is player $playerid$ aware of the fact that you have changed the state and remaining sequence of bomb $bombid$?

- Based on the observation and previous history, is player $playerid$ aware of the fact that you know a bomb phase has just been defused?

- Based on the observation and previous history, is player $playerid$ aware of the fact that you know the sequence of bomb $bombid$?