# OpenReview forum: "Theory of Mind for Multi-Agent Collaboration via Large Language Models"
_EMNLP/2023/Conference — EMNLP 2023 Main_

### Official Review · Reviewer_sosq · 2023-08-01

**Soundness:** 3

**Excitement:**

4: Strong: This paper deepens the understanding of some phenomenon or lowers the barriers to an existing research direction.

**Paper Topic And Main Contributions:**

This paper evaluated the ability of Large Language Models (LLMs) to carry out embodied interactions in a team-based task. The findings showed that LLM-based agents can perform complex multi-agent collaborative tasks with Theory of Mind (ToM) inference tasks on par with the current leading reinforcement learning algorithm.

**Questions For The Authors:**

1. Are the baseline models trained specifically for the task?
2. How does the rule-based text interface work and what does the provided feedback for error correction look like?
3. How many times the model will be evaluated on the task of bomb disaamble?
4. How much variation could be included in the task initial status?


**Reasons To Accept:**

The paper conducts Theory of Mind (ToM) evaluations via novel interactive team tasks, where agents' mental states dynamically evolve with each interaction. Also, the paper identifies systematic shortcomings that hinder the efficiency of Large Language Model (LLM)-based agents in collaboration and suggests a prompt-engineering approach to lessen these failures.

**Reasons To Reject:**

Details and background information are missing for the baseline reinforcement learning method MAPPO and the CBS Planner.

It seems unclear how the proposed novel collaborative task on bomb disaamble can be general enough to represent the model's ability to perform team interaction. More task analysis and the human baseline are needed.

**Reproducibility:**

4: Could mostly reproduce the results, but there may be some variation because of sample variance or minor variations in their interpretation of the protocol or method.

**Reviewer Confidence:**

2: Willing to defend my evaluation, but it is fairly likely that I missed some details, didn't understand some central points, or can't be sure about the novelty of the work.

---

> ### Author Rebuttal · Authors · 2023-08-25
>
> We thank the reviewer and address their comments individually below:
>
> Background and reproducibility details for MAPPO and CBS are provided in Section 5.2. Since the scope of our paper focuses on evaluating LLMs, we omit formal presentations and implementation details of RL and planning-based baselines. We will provide additional information in the camera-ready paper.
>
> We appreciate the reviewer's comments about generalizability and human baselines; we plan to explore them in future work as discussed in the Limitation section. Our task scenario is designed to represent real-world urban search and rescue tasks with components such as time pressure, information asymmetry, interdependent coordination, and decision-making under uncertainty. Artificial teams are expected to mimic foundational teamwork processes in human teams, including communication, backup behaviors, and shared mental model etc. We believe the scenario is a reasonable representation of general collaborative tasks with the potential to scale.
>
> Yes, the MARL algorithm is trained on the specific task environment for 45 million timestamps until convergence. CBS is a planning-based method without training but is also configured for the specific task.
>
> The text interface translates state observations into natural language descriptions and encodes LLM outputs into concrete action selections. Specifically, the text interface extracts state features from the game engine and replaces keywords in the templates. A typical observation description looks like: <Your observation is: Round: 1. Total team score: 0. Results: You inspected Bomb 1. This bomb is a 1-stage bomb, and its remaining sequence is Red. Room contents: You are currently in Room 0. Contents of this room include alpha, bravo, and charlie; Bomb 1. Teammate Locations: Player alpha is in Room 0; Player bravo is in Room 0; Player charlie is in Room 0. Communication messages sent by your teammates: Player alpha: "I am inspecting the bomb in Room 0. Bravo; please move to Room 3. Charlie; please move to Room 5.". Player bravo: "None". Player charlie: "None".>. Action encoding is done via keyword matching since LLMs are instructed to frame their responses in a certain format and structure. The text interface provides error feedback based on pre-programmed rules and message templates, such as <There is no bomb in the current location, Room {current_room}, for you to inspect.>.
>
> Teams of LLM-based agents were evaluated on the task scenario three times. Since the temperature parameter of LLMs is set to zero, their team behaviors are considerably consistent across trials. Performance of all experiment trials is presented in Table 1 with means and standard deviations.
>
> As introduced in Section 5.1, the task environment can be initialized with different starting locations, room connections, bomb distributions, and sequences.

---

### Official Review · Reviewer_tQsZ · 2023-08-02

**Soundness:** 4

**Excitement:**

4: Strong: This paper deepens the understanding of some phenomenon or lowers the barriers to an existing research direction.

**Justification For Ethical Concerns:**

The reproducibility of the paper may be affected because a large-scale model API provided by a commercial company was used.

**Missing References:**

The Theory of Mind section lacks many references to concurrent work in the field. I was able to easily find many paper about RL environments for Theory of Mind and testing LLM on Theory of Mind, such as "Improving Multi-Agent Cooperation using Theory of Mind" (https://arxiv.org/pdf/2007.15703.pdf), "Minding Language Models’ (Lack of) Theory of Mind: A Plug-and-Play Multi-Character Belief Tracker" (https://arxiv.org/pdf/2306.00924.pdf), and "Emergence of Theory of Mind Collaboration in Multiagent Systems" (https://arxiv.org/pdf/2110.00121.pdf).

**Paper Topic And Main Contributions:**

The paper is very interesting. It can be seen that LLM agents are able to explain their own intentions and understand the intentions of other agents when completing cooperative tasks. Among various LLM prompt engineering works, there are not many articles that have both interesting starting points and good storytelling.

**Questions For The Authors:**

I recommend to add a few case studies in the appendix to show a complete game process and give people a more intuitive impression.

**Reasons To Accept:**

Second-order ToM inference examines whether an agent can estimate the state of other agents, while third-order ToM inference examines whether an agent can guess what others think of themselves. These psychological settings are very interesting and are not limited to completing the tasks themselves.

**Reasons To Reject:**

The disadvantage of the article is that the scene is relatively single, and the scene is a fully shared information that only involves multi-agent cooperation. The human annotation results given in Table 2 did not explain the details and artificial judgment standards clearly. The implementation details of MAPPO implemented by the author were not explained.

**Reproducibility:**

4: Could mostly reproduce the results, but there may be some variation because of sample variance or minor variations in their interpretation of the protocol or method.

**Reviewer Confidence:**

3: Pretty sure, but there's a chance I missed something. Although I have a good feel for this area in general, I did not carefully check the paper's details, e.g., the math, experimental design, or novelty.

**Typos Grammar Style And Presentation Improvements:**

- Generally, papers related to RL will formalize their environment, including state space, observation space, action space, reward function, etc. This paper focuses mostly on prompt engineering, it is not necessary to use formulas to describe these things. But, it is strongly recommended to describe the RL environment in a bullet-point list.
- It is suggested to put the introduction of “bombs” and “rewards” from lines 294-310 into the introduction of the environment in Section 3.1.
- The two embedded images by the author are too blurry. It is recommended to save them as PDF and then embed them in LaTeX.
- Paragraph format at line 498.

---

> ### Author Rebuttal · Authors · 2023-08-25
>
> We thank the reviewer for their appreciation of our paper.
>
> We agree with the reviewer's desire to apply our method to more complicated tasks or competitive scenes and plan to do so in future work. However, the judgment that our task scenario is "single" and involves "fully shared information" is slightly inaccurate. The interaction history between agents and the environment is not shared within the team. Each agent has partial observability of its own room, plus the communication messages selectively sent by other agents. Given that leading MARL algorithm such as MAPPO can not solve the task optimally, we consider it as a relatively challenging teamwork scenario.
>
> The reviewer's comments about human annotation are appreciated. More details about human annotations can be found in Section 6.5.2 and the second paragraph of Limitations. Specifically, human annotators were instructed to judge LLM's ToM inference questions based on the following principles: 1) if the target agent is present in the current room and observes the belief change, 2) if the target agent has been to this room before, 3) if the belief change has been communicated to the target agent. It is worth mentioning that high-order ToM inferences involving communication are naturally ambiguous. These corner cases were discussed among annotators to ensure a consistent standard across conditions.
>
> Reproducibility details about MAPPO are provided in Section 5.2. Since the algorithm is open-sourced and we used the default hyperparameters from the original paper, we did not include further implementation details due to page restrictions. We will provide additional information in the camera-ready paper.
>
> We appreciate the reviewer's comments on case studies, potential references, and presentation improvements and will address them in the final paper.

---

### Official Review · Reviewer_Bhvf · 2023-08-05

**Typos Grammar Style And Presentation Improvements:** Ln 498
**Soundness:** 4

**Excitement:**

4: Strong: This paper deepens the understanding of some phenomenon or lowers the barriers to an existing research direction.

**Paper Topic And Main Contributions:**


The paper evaluated large language models on their abilities in multi-agent collaborations. It conducted the evaluation in the MARL task environment where three LLMs play the role of three agents to safely defuse bombs. The game requires communication to exchange partial information and complete the tasks collaboratively. The paper engineered prompts to translate the game environment into natural language, a list of 4 actions to select, belief state, and a request for communication messages. The paper conducted comparisons among 6 variations of LLM, reinforcement learning, and rule based methods. The paper identified key failure points of the models and proposed future directions.


**Reasons To Accept:**

The paper proposed a novel evaluation of the large language models on a multi-agent collaboration game. It conducted thorough and sophisticated experiments to benchmark 6 methods with the task environment. It evaluates three orders of theory of mind capabilities of the models with interesting findings. The paper identified systematic failures of the LLM based agents and proposed ideas to mitigate the issues in current and future works. The paper is rigorous and well-written.


**Reasons To Reject:**

It is unclear how ToM inference tasks are evaluated. The description is fairly concise without enough information to conclude the claim about theory of mind inferences.

The action space (3) of the agents is relatively small. Would be helpful to discuss ideas on how this work could be applied in a larger scale.


**Reproducibility:**

3: Could reproduce the results with some difficulty. The settings of parameters are underspecified or subjectively determined; the training/evaluation data are not widely available.

**Reviewer Confidence:**

4: Quite sure. I tried to check the important points carefully. It's unlikely, though conceivable, that I missed something that should affect my ratings.

---

> ### Author Rebuttal · Authors · 2023-08-25
>
> We thank the reviewer for acknowledging our contributions to the field. But we seek to clarify some details of our methodology here.
>
> The reviewer's comments about ToM inference evaluation are well-taken. The design principle of the ToM inference task is inspired by the Sally–Anne test, the most widely used ToM task in human studies. Every time an agent conducts an action, we pose a belief reasoning question, asking if another agent (i.e., target agent) is aware of the potential consequence of this action. The consequence here can be either a state change (e.g., a bomb has been defused) or a belief change (e.g., Alpha has explored Room 5 and found Bomb 3 in the room). An agent equipped with ToM should realize that while they know the consequence, the target agent might hold a false belief about it. To evaluate whether LLM-based agents answer these questions correctly, human annotators were hired to provide subjective judgment based on fully observable interaction and communication history. Specifically, the following principles are considered: 1) if the target agent is present in the current room and observes the consequence, 2) if the target agent has been to this room before, 3) if the consequence has been communicated to the target agent. Additionally, we plan to explore other evaluations of LLM-based agent's ToM in future work, as discussed in Section 6.5.2 and Limitations.
>
> We appreciate the reviewer's comments about the scalability of our work but would like to make a clarification. The size of the action space for our test environment is actually 9. The agent can choose to go to any of the 5 rooms, use wire-cutters in 3 colors, or inspect the bomb in the current room. We plan to evaluate our method in more complicated environments in future work. LLM-based agents might struggle in tasks where the state and action space are hard to describe in natural language (e.g., continuous space). Potential solutions to enable LLMs' embodied interactions in those tasks include combining LLM with low-level planners or using prompt engineering to divide tasks into smaller subtasks.

---

### Meta-Review · Area_Chair_zQJS · 2023-09-15

**Recommendation:** 5

**Metareview:**

Reviews for this paper were quite consistently positive, with scores 4,4,3 (soundness) and 4,4,4 (excitement).

Condensing the reviews, the following were among the strengths and weaknesses mentioned:

Strengths:

- The reviewers consistently found the paper well-written, and the results interesting.
- Highlights systematic failures of LLM based agents (R1, R3)
- interesting psychological settings (R2)

Weaknesses:

- some lack of clarity on implemention/evaluation (R1, R2, R3)
- small action space (R1) and limited settings (R2, R3)

---

### Decision · Program_Chairs · 2023-10-07

**Decision:**

Accept-Main

**Comment:**

Reviews for this paper were quite consistently positive, with scores 4,4,3 (soundness) and 4,4,4 (excitement).

Condensing the reviews, the following were among the strengths and weaknesses mentioned:

Strengths:

- The reviewers consistently found the paper well-written, and the results interesting.
- Highlights systematic failures of LLM based agents (R1, R3)
- interesting psychological settings (R2)

Weaknesses:

- some lack of clarity on implemention/evaluation (R1, R2, R3)
- small action space (R1) and limited settings (R2, R3)